# Effect of Hydrocortisone on Angiotensinogen (*AGT*) Mutation–Causing Autosomal Recessive Renal Tubular Dysgenesis

**DOI:** 10.3390/cells10040782

**Published:** 2021-04-01

**Authors:** Min-Hua Tseng, Shih-Ming Huang, Martin Konrad, Jing-Long Huang, Steven W. Shaw, Ya-Chung Tian, Ho-Yen Chueh, Wen-Lang Fan, Tai-Wei Wu, Jhao-Jhuang Ding, Ming-Chou Chiang, Shih-Hua Lin

**Affiliations:** 1Division of Nephrology, Department of Pediatrics, Chang Gung Memorial Hospital and Chang Gung University, Taoyuan 330, Taiwan; doc31089@gmail.com; 2Department of Biochemistry, National Defense Medical Center, Taipei 114, Taiwan; shihming@ndmctsgh.edu.tw; 3Department of General Pediatrics, University Children’s Hospital Münster, 481 Münster, Germany; konradma@uni-muenster.de; 4Division of Pediatric Allergy, Asthma, and Rheumatology, Department of Pediatrics, Chang Gung Memorial Hospital and Chang Gung University, Taoyuan 330, Taiwan; long@adm.cgmh.org.tw; 5Department of Obstetrics and Gynecology, Taipei Chang Gung Memorial Hospital and Chang Gung University, Taipei 114, Taiwan; doctor.obsgyn@gmail.com; 6Division of Nephrology, Department of Medicine, Chang Gung Memorial Hospital and Chang Gung University, Taoyuan 330, Taiwan; dryctian@cgmh.org.tw; 7Department of Obstetrics and Gynecology, Chang Gung Memorial Hospital and Chang Gung University, Taoyuan 330, Taiwan; doc31089@yahoo.com.tw; 8Genomic Medicine Core Laboratory, Chang Gung Memorial Hospital, Taoyuan 333, Taiwan; alangfan@gmail.com; 9Fetal and Neonatal Institute, Division of Neonatology Children’s Hospital Los Angeles, Department of Pediatrics, Keck School of Medicine, University of Southern California, Los Angeles, CA 900, USA; wutcz8@gmail.com; 10Department of Pediatrics, Tri-Service General Hospital, Taipei 114, Taiwan; jamesdin1124@gmail.com; 11Division of Neonatology, Department of Pediatrics, Chang Gung Memorial Hospital and Chang Gung University, Taoyuan 330, Taiwan; newborntw@gmail.com; 12Division of Nephrology, Department of Medicine, Tri-Service General Hospital, Taipei 114, Taiwan

**Keywords:** renal tubular dysgenesis, angiotensinogen, rescue therapy, founder effect

## Abstract

We has identified a founder homozygous E3_E4 del: 2870 bp deletion + 9 bp insertion in *AGT* gene encoding angiotensinogen responsible for autosomal recessive renal tubular dysgenesis (ARRTD) with nearly-fatal outcome. High-dose hydrocortisone therapy successfully rescued one patient with an increased serum Angiotensinogen (AGT), Ang I, and Ang II levels. The pathogenesis of ARRTD caused by this *AGT* mutation and the potential therapeutic effect of hydrocortisone were examined by in vitro functional studies. The expression of this truncated AGT protein was relatively low with a dose-dependent manner. This truncated mutation diminished the interaction between mutant AGT and renin. The truncated AGT also altered the glucocorticoid receptor (GR)-dependent transactivation, indicating that AGT may affect the development of proximal convoluted tubule by alteration of glucocorticoid-dependent transactivation. In hepatocytes, hydrocortisone increased the AGT level by accentuating the stability of mutant AGT and increasing its binding with renin. Therefore, hydrocortisone may exert the therapeutic effect through the enhanced stability and interaction with renin of truncated AGT in patients carrying this AGT mutation.

## 1. Introduction

Autosomal recessive renal tubular dysgenesis (ARRTD) featured by the absence or poor-differentiation of proximal convoluted tubules on histology, maternal oligohydramnios, pulmonary hypoplasia, profound hypotension, and anuria after birth is a rare inherited disorder caused by the inactivating mutations of genes responsible for renin-angiotensin system (RAS) [1,2,3,4,5,6]. To date, more than 80 different mutations in genes encoding proteins of RAS have been identified in patients with ARRTD. The majority of mutated genes are *ACE* followed by *REN*, *AGT* and *AGTR1* [6,7]. Almost all affected fetuses die either in uterus or after birth with refractory hypotension and/or respiratory failure [7,8,9,10]. Although the compromised renal perfusion caused by the defect in RAS has been proposed to be responsible for the development of renal tubular dysgenesis, the exact pathogenesis remains to be elucidated [11,12,13]. Until now, there was no definitive treatment for patients with ARRTD. It is crucial to develop specific and recuse therapy.

Recently, we have reported six patients with ARRTD caused by *AGT* (angiotensinogen) mutation from six unrelated families [14]. This homozygous E3_E4 del: 2870 bp deletion + 9 bp insertion in *AGT* resulted in the skipping of exons and the generation of truncated protein (1–292 amino acids) as well as diminished serum AGT, Ang I, and Ang II. The rapidly fatal course even with aggressive therapy including inotropic agents, plasma infusion, and peritoneal dialysis was notable in 5 of 6 patients. Based on the previous studies showing that the glucocorticoid acted as a transcription regulator of *AGT*, the administration of persistently higher dose of hydrocortisone achieved a better blood pressure response and rescued one patient without the need for dialysis. Since no ARRTD patients with the truncating variant have survived to date, the spontaneous recovery of hypotension and renal hypoperfusion is unlikely. To the best of our knowledge, this is the first successful rescue therapy of ARRTD. Nevertheless, the molecular mechanism of the potential rescue effect of hydrocortisone on ARRTD remains elusive. In this study, we explored the pathogenesis of this *AGT* mutation and elucidated the potential rescue role of hydrocortisone in vitro functional analysis.

## 2. Materials and Methods

### 2.1. Index Case

This study was approved by the ethics committee on human studies at Chang Gung Memorial Hospital in Taiwan (IRB 201902035A3). An index neonate with ARRTD caused by homozygous E3_E4 del: 2870 bp deletion + 9 bp insertion in *AGT* was successfully rescued by systemic hydrocortisone. This mutation led to the exclusion of exon 3 and 4, and generated the truncated AGT (1–292 amino acids). A potentially candidate binding motif (LQDLL) of nuclear receptor was excluded by this mutation (Figure 1) [14].

### 2.2. Construction of Plasmids

Various AGT coding regions were synthesized by polymerase chain reaction and subcloned into EcoRI and XhoI sites of vector pSG5.HA [15]. Three pSG.HA.AGTs, wild type, truncated 1–292, and 1–375 (R375Q), were created. The pSG.HA.AGT (1–375) was used for negative control due to the recurrent and truncating mutation. Plasmid quality has been checked by GE NanoVue spectrophotometer (GE Healthcare Systems, Chicago, IL, USA). A A260/A280 value of 1.80–1.90 was an indication for pure plasmid DNA. All plasmid DNAs were verified by sequencing analysis (Mission Biotech, Taipei, Taiwan).

### 2.3. Cell Culture, Transfection and Protein Stability Assay

HK-2 (human kidney-2) cells purchased from the Bioresource Collection and Research Center were grown in Dulbecco’s Mod. of Eagle’s Medium/Ham’s F-12 medium with 10% fetal bovine serum. L02 cells were grown in Dulbecco’s modified Eagle’s medium supplemented with 10% fetal bovine serum. HK-2 and L02 cells (Cell Bank of China Science, Shanghai, China) were transfected with indicated amount of pSG5.HA.AGT–wild type and pSG5.HA.AGT (1–292 and 1–375/R375Q) constructs. For protein stability analysis, transfected cells were treated with 50 μg/mL cycloheximide (CHX) for 20, 40, 80, 180, and 360 min. The cells were treated with vehicle or respective hormones for 24 h and were harvested for Western blotting analysis.

### 2.4. Western Blotting

The cell lysates were prepared in lysis buffer (100 mM Tris-HCl (pH 8.0), 150 mM NaCl, 0.1% SDS, 1% Triton X-100) at 4 °C, separated by sodium dodecyl sulfate polyacrylamide gel electrophoresis, and electrotransferred onto polyvinylidene difluoride membranes (Millipore, Bedford, MA). The following proteins were detected using specific antibodies: Anti-HA (clone 3F10, Roche, 1:5000 dilution), Anti-α-actinin (H-2, sc-17829, Santa Cruz Biotechnology, 1:5000 dilution), Anti-AGT (11992-1-AP, Proteintech Group, 1:1000 dilution), Anti-GR (H-300, sc-8992, Santa Cruz Biotechnology, 1:1000 dilution), and Anti-p53 (DO-1, sc-126, Santa Cruz Biotechnology, 1:1000 dilution).

### 2.5. Proximity Ligation Assay (PLA)

Cells in situ PLA signal were measured by Duolink^®^ In Situ Fluorescence Kit (Sigma-Aldrich, Saint Louis, MO, USA) according to the manufacturer’s instructions. In brief, HK-2 and L02 cells were cultured on coverslips in DMEM/F-12 with 10% FBS and DMEM medium with 10% FBS, respectively. These cells were transfected with pSG5.HA vector, HA-tag hAGT (full-length, a.a. 1–485), HA-tag hAGT (truncated, a.a. 1–292), and HA-tag hAGT mutant (R375Q) constructs. Primary antibodies, anti-Renin (rabbit polyclonal, HPA005131, Sigma), and anti-AGT (mouse monoclonal, 60126–1-Ig, Proteintech Group) were mix and dilute at 1:100 in the Duolink^®^ Antibody Diluent (1×). For obtaining the digital images, the samples were analyzed by fluorescence microscope.

### 2.6. Luciferase Reporter Analysis of Glucocorticoid Receptor (GR)-Dependent Transactivation

The expression pKSX vectors for human GR and reporter gene for the MMTV-LUC have been described previously [16]. HK-2 cells were plated in 24-well plates and transfected using jetPEI (Polyplus Transfection Inc., New York, NY, USA) following the manufacturer’s protocol (Promega luciferase assay kit and DLR2 model). The total DNA was adjusted to 1.0 μg using empty pSG5.HA vector. Cells were harvested for luciferase reporter assays using a Promega Luciferase Assay Kit (Madison, WI, USA). Values are expressed numerically as relative light units. Luciferase activity is presented as the mean ± SD of two transfected wells and is representative of at least three independent experiments. The results shown are representative of at least three independent experiments.

### 2.7. Statistical Analysis

Resulting mean data were compared with a Student two-tailed t test. The percent relative standard deviation (SD), was calculated as 100 multiplied by SD divided by the mean.

## 3. Results

### 3.1. In Vitro Expression of Truncated AGT

The expressions of wild-type AGT (amino acids (aa) 1–485) and two truncated AGT (AA 1–292 and 1–375/R375Q) proteins were examined after transient transfection of AGT in an immortalized proximal tubule epithelial HK-2 cell line. HK-2 cells were transiently transfected with the indicated amount of indicated pSG5.HA.AGT plasmid DNAs and incubated for 46 h. The cell lysates were subjected to Western blotting with antibodies against angiotensinogen (AGT). Alpha-actinin (ACTN) was used as the loading control. The expressions of wild-type AGT and two truncated AGT proteins were in a dose-dependent manner with a relatively low expression of truncated AGT (AA 1–292) (Figure 2A). To further examine their relative stabilities, the HK-2 cells, a proximal tubular cell line derived from normal kidney, were transiently transfected with 0.4 μg pSG5.HA.AGT (wild-type and amino acids 1–375/R375Q) and 0.8 μg pSG5.HA.AGT (amino acids 1–292) and transfected cells were treated with 50 μg/mL cycloheximide (CHX), a *de novo* protein synthesis inhibitor, for 20, 40, 80, 180, and 360 min and analyzed in the Western blot analysis for the quantitation of these AGT proteins with ImageJ software. Compared to wild-type AGT, the truncated AGT (AA 1–292) protein was more stable at the longer CHX treatment time, but the truncated AGT (AA 1–375/R375Q) protein was less stable at the shorter CHX treatment time (Figure 2B). The relative stability was calculated and compared with 0 h CHX treatment (Figure 2C).

### 3.2. Measurement of Renin-AGT Interaction in Kidney

Due to this deletion mutation of *AGT* resulting in skipping of exons, we further evaluated whether this truncated AGT impaired its interaction with renin in kidney. HK-2 cells were transiently transfected with 0.5 μg pSG5.HA vector and indicated pSG5.HA.AGT plasmid DNAs. Using the proximity ligation assay (PLA), we examined the possible interaction defect between renin and wild-type and truncated AGT proteins. We analyzed the samples with a fluorescence microscope and obtained digital images. As shown in Figure 3, there was a significant decrease in number of PLA of two truncated AGT proteins (AA 1–292 and 1–375/R375Q), but not wild-type protein, indicating that this truncated AGT protein might diminish the AGT-Renin interaction.

### 3.3. Effect of Hydrocortisone on the Expression of AGT in Liver

Because liver is the major organ for the AGT synthesis, we used a normal human hepatic cell line L02 to examine whether hydrocortisone could increase the endogenous expression of AGT. The L02 cells were treated with indicted amount of hydrocortisone and dexamethasone for 24 h. The cell lysates were subjected to Western blotting with antibodies against AGT and GR. ACTN was used as the loading control. By increasing concentrations of hydrocortisone and dexamethasone, a decrease in glucocorticoid receptor (GR) with both drugs was observed (Figure 4A). The ubiquitin-proteasome pathway in regulating GR protein turnover was a classic mechanism to terminate glucocorticoid responses [17]. There was no apparent change of endogenous AGT proteins treated with hydrocortisone and dexamethasone in L02 cells. The Western blotting data showed that there was no apparent change of these three exogenous AGT proteins treated with either dexamethasone (100 nM) or hydrocortisone (10 μM) for 24 h in L02 cells transiently transfected by 0.4 μg pSG5.HA.AGT wild-type (AA 1–485) and pSG5.HA.AGT (AA 1–375/R375Q) and 0.8 μg pSG5.HA.AGT truncated (AA 1–292) (Figure 4B). Lastly, we checked whether hydrocortisone could differentially affect the protein stability of wild-type AGT and truncated AGT using the CHX pulse-chase experiment. The truncated AGT (AA 1–292) was more stable than wild-type AGT in L02 cells. With a 46-h hydrocortisone treatment, our quantitative data demonstrated that hydrocortisone increased the amount of exogenous wild-type and truncated AGT (AA 1–292) proteins (Figure 4C,D compare the absence and presence hydrocortisone at 0 h CHX treatment time) and the stability of truncated AGT (AA 1–292) was better than wild-type AGT protein after the 90-min CHX treatment (Figure 4C,D). In Figure 4E, pretreated 2 h proteasome inhibitor MG132 demonstrated that exogenous HA.AGT (AA 1–485 and 1–292) or endogenous p53 could be rescued from the protein degradation pathway in the 120-min CHX treatment of L02 cells. This indicates that both wild-type and truncated AGT could be degraded by proteasome. These findings implicated that the increased serum AGT by hydrocortisone did not result from the direct increase of its production, but the stabilization of cytosolic truncated AGT.

### 3.4. Effect of Hydrocortisone on the Interaction between Renin and Mutant AGT in Liver

Liver is the major organ for the AGT synthesis. The PLA study showed that hydrocortisone enhanced AGT-Renin interaction in both wild type and truncated AGT in a normal human hepatic L02 cell line (Figure 5A,B). This pinpointed that hydrocortisone could enhance the interaction between renin and AGT by increasing the stability of mutant AGT and may contribute the subsequent increase of Ang I and II.

### 3.5. Mutant AGT Impaired the Glucocorticoid Receptor (GR)-Dependent Transactivation

Previous studies showed that GR might play important roles in the development of proximal convoluted tubule [18,19]. In addition, the truncated AGT identified in our case excluded the potential binding motif of nuclear receptor. Here, we addressed whether full-length AGT could be involved in the GR-dependent transactivation and mutant AGT could alter this GR-dependent transactivation. HK-2 cells were transfected with 0.25 μg of the MMTV-LUC reporter plasmid and pKSX.GR (0.15 μg) and/or indicated amounts of various AGT expression vectors in the absence or presence of 100 nM Dexamethasone or 1 μM hydrocortisone for 46 h. We first observed that dexamethasone and hydrocortisone were able to activate GR-dependent MMTV-Luc reporter activities. Wild-type AGT enhanced GR-dependent MMTV-Luc reporter activities in below 180 ng for dexamethasone stimulation and 40 ng for hydrocortisone stimulation, whereas suppressed GR-dependent MMTV-Luc reporter activities over these critical amounts. Both truncated AGTs (1–292 and 1–375/R375Q) suppressed this GR-dependent MMTV-Luc reporter activity in all tested dosages (Figure 6A). These results suggested that truncated AGT may affect the development of proximal convoluted tubule by alteration of GR-dependent transactivation (Figure 6B).

## 4. Discussion

In this study, we performed the functional analysis of a homozygous deletion mutation of *AGT* responsible for ARRTD. In in vitro studies, this AGT mutant had a lower expression level than wild type AGT. PLA study demonstrated the attenuated interaction of this truncated protein with renin. Hydrocortisone could not only enhance the stability of truncated AGT but also accentuate its interaction with renin (Figure 7).

Despite more than 150 patients with ARRTD caused by different mutations in genes encoding proteins of RAS [1,2,18,19,20], only few patients with ARRTD from unrelated families caused by *AGT* mutations have been reported. Seven different mutations including 5 missense, 1 nonsense, and 1 large deletion mutations on *AGT* were identified. All of the mutants were located in the serpine domain [1,8,14,21]. Of note, the serpine domain is critical for the AGT cleavage by renin [18,22,23]. The present case with defect in *AGT* resulting in the deletion of serpine domain had low serum Ang I and II levels. By PLA analysis, we demonstrated this truncated AGT identified in our patient impaired the interaction between renin and mutant AGT. Thus, this genetic defect of *AGT* led to diminished serum AGT, and the low Ang I and Ang II results from, at least partially, the impaired AGT cleavage by renin. The vitro study revealed low expression of mutant AGT in line with the findings of our recent report that low amount of truncated AGT protein in liver and kidneys of patients with same *AGT* mutation [13]. Consistent with our finding, low serum AGT has been reported in cases with other truncated mutation, R375Q [24,25,26]. This attenuated expression of truncated AGT protein could be caused by either reduced hepatic production or enhanced hepatic degradation of truncated AGT.

Organic hypoperfusion, especially renal hypoperfusion in fetus, resulting from dysregulation of RAS is the main pathogenesis of renal tubular dysgenesis [6,9,12]. No correlation between clinical severity and mutated genes pinpointed that the functional RAS was critical for renal tubular development and the low angiotensin production was the bona fide cause of organic hypoperfusion. We found that this *AGT* mutation led to truncated AGT that excluded the amino acids that contained the potential binding motif (LQDLL) for nuclear receptor [27]. Since GR, one of nuclear receptors, is abundant in proximal convoluted tubule during the tubular development [18,19,28]. In mouse and rat, the AGT is primarily detected in proximal tubules at early gestation [29,30,31,32]. Therefore, the AGT may play an important role in acting as a growth factor during proximal convoluted tubule. In line with this speculation, we found that this mutant AGT could alter the GR-dependent transactivation and might provide another pathogenesis of renal tubular dysgenesis caused by this *AGT* mutation.

Glucocorticoid has been shown to be the regulators of generation of AGT in liver by increasing the abundance mRNA of *AGT* in vivo and in vitro studies [15,33,34]. Our in vitro study showed that the effect of hydrocortisone and dexamethasone did not affect the production of truncated AGT. However, hydrocortisone was shown to enhance the stability of the truncated AGT with the CHX treatment. This enhanced stability of truncated AGT may allow it more interaction with renin. As demonstrated by PLA, the hydrocortisone not only actually increased the interaction of the truncated AGT but also wild-type AGT with renin. This interesting finding may suggest that the beneficial effect of hydrocortisone is not limited to this specific mutation. In addition, hydrocortisone has been demonstrated to exert an effect on the expression of GR, which is abundant in proximal convoluted tubule during the development [19,31,35]. To elucidate this potential benefit of hydrocortisone on GR expression on the development of proximal convoluted tubule, further animal study is warranted.

## 5. Conclusions

This large deletion of *AGT* identified in the index case led to the truncated AGT and decreased its cleavage by renin with subsequent low Ang I and Ang II generation. This truncated AGT also altered the GR-dependent transactivation. High dose of hydrocortisone, at least in part through increasing the stability of the truncated AGT and enhancing the cleavage of AGT by renin, may be a potential therapy for ARRTD caused by *AGT* mutation.

## Figures and Tables

**Figure 1 cells-10-00782-f001:**
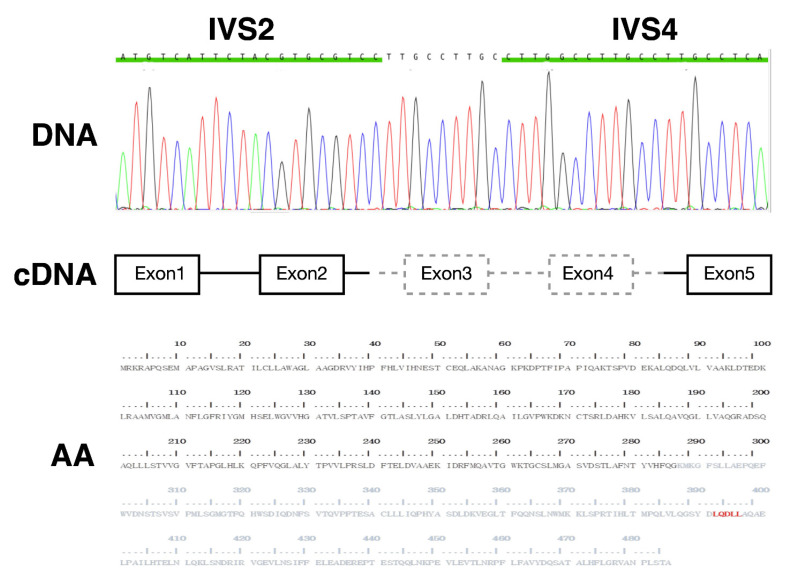
**Sanger sequence, cDNA, and amino acids sequence**. Sanger sequencing illustrated a large deletion of Angiotensinogen (AGT), E3_E4 del: 2870 bp deletion with 9 bp insertion. With cDNA analysis, this deletion of AGT leads to a truncated protein (292 amino acids).

**Figure 2 cells-10-00782-f002:**
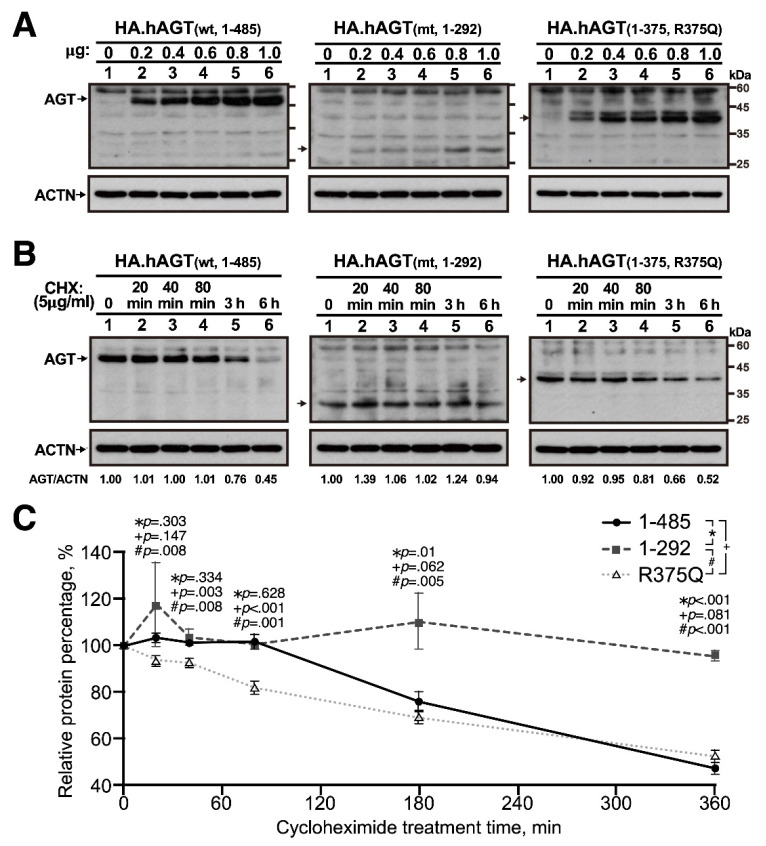
**The expression level and stability of various AGT proteins.** (**A**) Human kidney-2 (HK-2) cells (5 × 10^4^ cells/well) were transiently transfected with HA.AGT wild-type (AA 1–485), pSG5.HA.AGT (AA 1–375/R375Q), and pSG5.HA.AGT truncated (AA 1–292). (**B**,**C**) HK-2 cells (5 × 10^4^ cells/well) were transiently transfected with 0.4 μg pSG5.HA.AGT wild-type (AA 1–485) and pSG5.HA.AGT (AA 1–375/R375Q) and 0.8 μg pSG5.HA.AGT truncated (AA 1–292). (**C**) The relative stability was calculated and compared with 0 h cycloheximide (CHX) treatment. The results (**A**,**B**) are representative of three independent experiments.

**Figure 3 cells-10-00782-f003:**
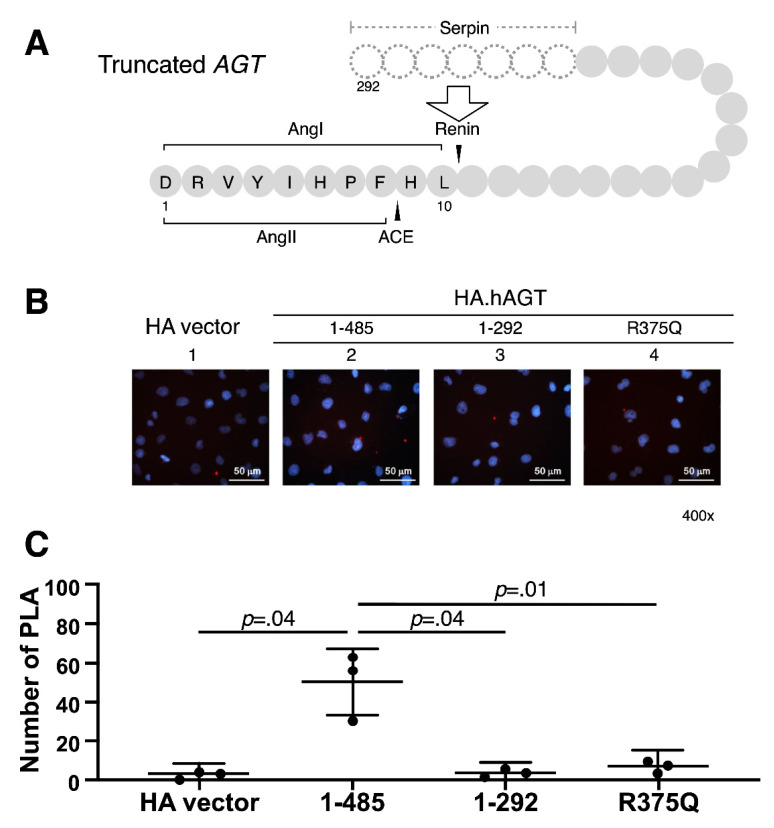
**Proximity ligation assay between AGT and renin in kidney.** (**A**) Illustration of deleted region of truncated AGT protein involving serpin domain and cleavage sites of renin and angiotensin-converting enzyme (ACE); (**B**,**C**) HK-2 cells (5 × 10^4^ cells/well) were transiently transfected with 0.4 μg pSG5.HA.AGT wild-type (AA 1–485) and pSG5.HA.AGT (AA 1–375/R375Q) and 0.8 μg pSG5.HA.AGT truncated (AA 1–292). Measurement of Renin-AGT interaction by proximity ligation assay. The results (**C**) are representative of three independent experiments.

**Figure 4 cells-10-00782-f004:**
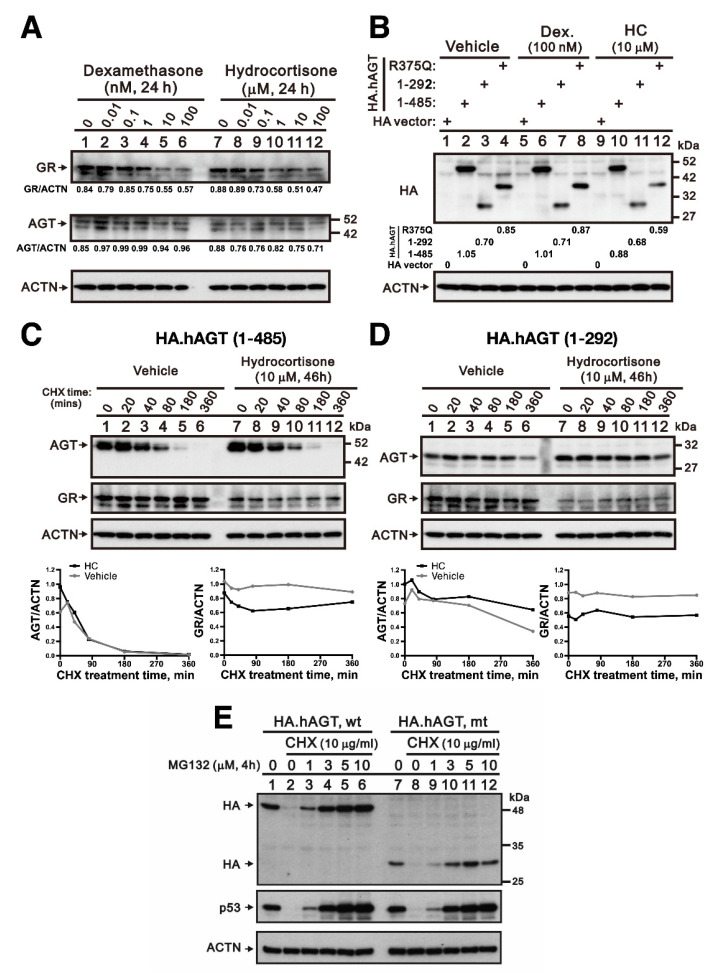
**The effect of hydrocortisone on mutant AGT proteins.** (**A**) L02 cells were treated with indicted amount of hydrocortisone and dexamethasone for 24 h; (**B**) L02 cells were transiently transfected with 0.4 μg pSG5.HA.AGT wild-type (AA 1–485) and pSG5.HA.AGT (AA 1–375/R375Q) and 0.8 μg pSG5.HA.AGT truncated (AA 1–292) and transfected cells were treated with dexamethasone and hydrocortisone for 24 h; (**C**,**D**) L02 cells were transiently transfected with pSG5.HA.AGT (1–485 and 1–292) and transfected cells treated with 10 μM hydrocortisone for 46 h and 50 μg/mL CHX for 20, 40, 80, 180, and 360 min. The relative stabilities of wild-type and mutant AGT proteins were calculated and compared with 0 h CHX treatment. Both wild-type and truncated AGTs increased with a decreased glucocorticoid receptor (GR) after hydrocortisone treatment. (**E**) Effect of proteasome inhibitor on AGT degradation. L02 cells were transiently transfected with 0.4 μg pSG5.HA.AGT wild-type (AA 1–485) and 0.8 μg pSG5.HA.AGT truncated (AA 1–292) and transfected cells treated with 0, 1, 3, 5, and 10 μM MG132 for 2 h and then treated with 10 μg/mL CHX for 120 min. p53 is for a control MG132 treatment and alpha-actinin (ACTN) is for a loading control. The results (**A**–**E**) are representative of three independent experiments.

**Figure 5 cells-10-00782-f005:**
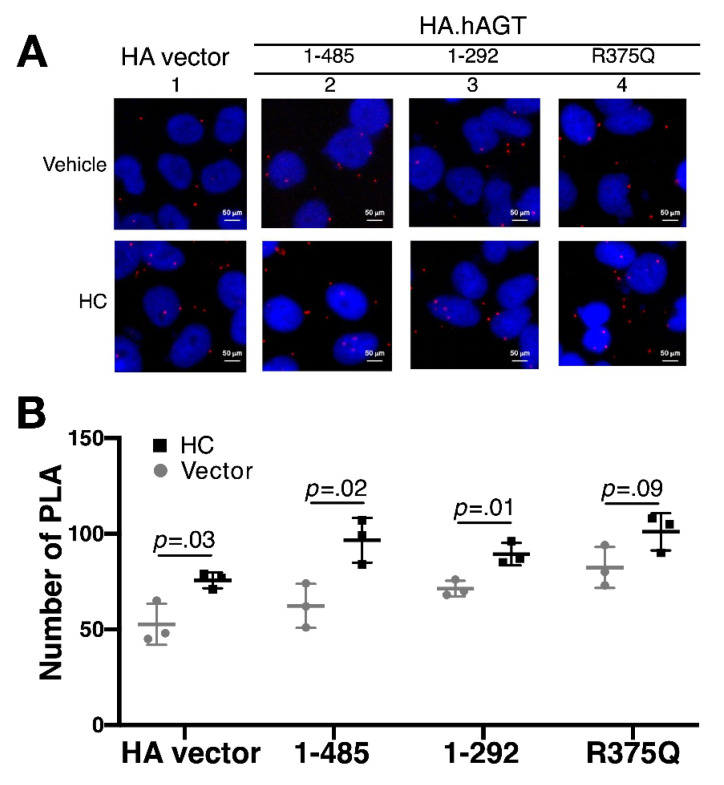
**The effect of hydrocortisone on the interaction between AGT proteins and renin in liver.** (**A**) L02 cells (8 × 10^4^ cells/well) were transiently transfected with 0.4 μg pSG5.HA.AGT wild-type (AA 1–485) and pSG5.HA.AGT (AA 1–375/R375Q) and 0.8 μg pSG5.HA.AGT truncated (AA 1–292) and transfected cells were treated with hydrocortisone for 24 h. (**B**) Measurement of Renin-AGT interaction by proximity ligation assay. The results are representative of three independent experiments.

**Figure 6 cells-10-00782-f006:**
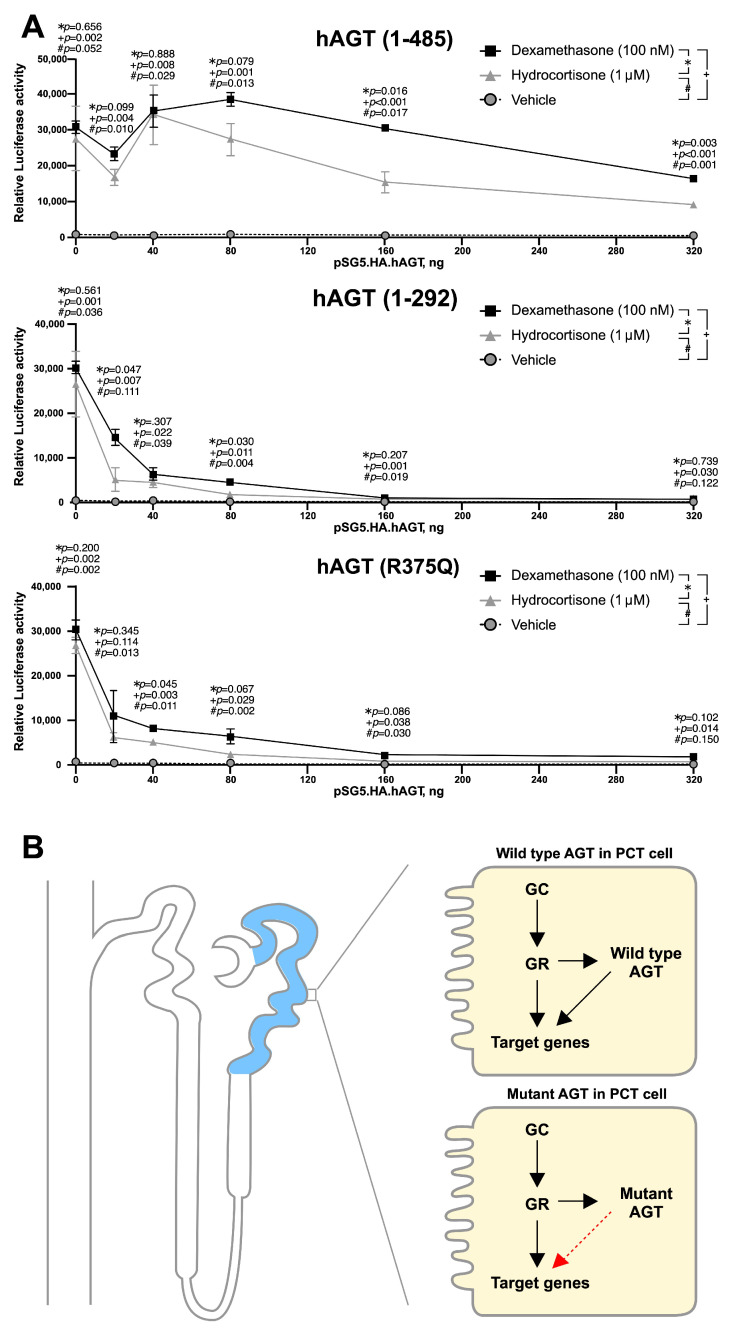
**The functional status of various AGTs examined by the modulation of GR-dependent MMTV-LUC activity in HK-2 cells.** (**A**) HK-2 cells were transfected with 0.25 μg of the MMTV-LUC reporter plasmid and pKSX.GR (0.15 μg) and/or indicated amounts of various AGT expression vectors in the absence or presence of 100 nM Dexamethasone or 1 μM hydrocortisone for 46 h. The presented data are the means of three experiments (mean ± S.D.; *n* = 3); (**B**) The mutant AGT exerts different glucocorticoid receptor-dependent transactivation than wild AGT.

**Figure 7 cells-10-00782-f007:**
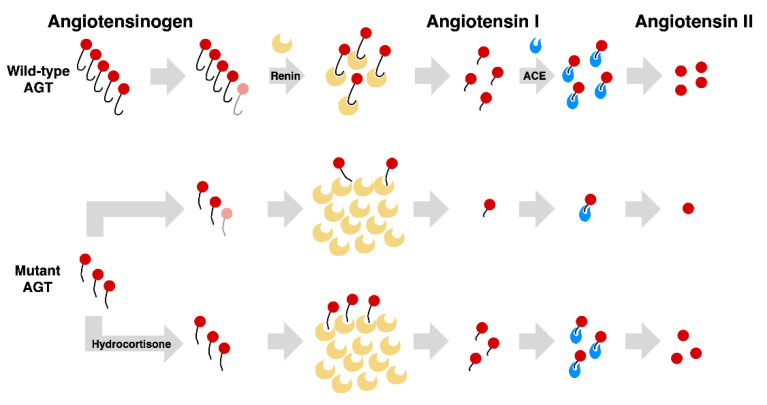
**Proposed Pathogenesis of Angiotensin II deficiency due to large deletion of *AGT* in Rental Tubular Dysgenesis.** Normally, wild-type AGTs are converted to angiotensin II through enzymes of renin and ACE. Mutation in *AGT* gene yields a shortened and abnormal AGT with decreased cleavage by renin and subsequent low serum angiotensin I and II. Hydrocortisone increases the production of angiotensin II by enhancing the interaction between renin and truncated AGT.

## Data Availability

The data supporting reported results can be found at https://livendmctsghedu-my.sharepoint.com/:x:/g/personal/jamesdin1124_office365_ndmctsgh_edu_tw/ETfx47Rl7NFDtxvfYpzhfsEBs5CFg_MaIzEyIqfb0Vk1Dg?e=wp3049, accessed on 31 March 2021.

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
