# Peer review of "Effect of Hydrocortisone on Angiotensinogen (AGT) Mutation–Causing Autosomal Recessive Renal Tubular Dysgenesis"

_cells, 2021, doi:10.3390/cells10040782_

Round 1
Reviewer 1 Report
The authors found a mutation in human angiotensinogen associated with the autosomal recessive renal tubular 30 dysgeneses (ARRTD). In this sequel, the authors study how this mutation plays a role in developing the ARRTD and show how Hydrocortisone can be beneficial.
In other to other groups to repeat the experiment, the authors had to give more details about their experimentation.
How the authors verified the quality of the plasmids?
Where were the cells purchased? How long the plasmid was transfected, 12 hours, 24 hours?
What is the composition of the lysed buffer? Any protease inhibitor?
Where the antibodies were purchased, and their working concentration?
Why the authors describe coefficient 123 of variance (COV) when they do not use this parameter in the text?
In section 2.3, the authors wrote cycloheximide for 46 h, while in Fig.2, then use 0 to 3 hours.
In the PLA assay, the authors use a different cell culture medium described in the cell culture section.
Figure 1 is roughly the same as the previously published content. Is there any new information?
In Fig. 2 A, the authors show no bands in the absence of Wild type construct. However, a bar similar to the wt construct appears in both mutant constructs, suggesting endogenous AGT detection. Do the authors use HA antibody or AGT antibody, or both?
The authors suggest that HA-hAGT 1-292 is more stable than other constructs. We can imagine that the AGT reduction in the cells is mostly due to AGT being secreted from the cells because we do not observe any degradation products. Since the HA-hAGT 1-292 is less expressed than wild type constructs, it suggests a problem in folding in RE and Golgi that slow down its release in secret pathways. To confirm the AGT's degradation, authors should perform this experiment in the presence of protein degradation inhibitors.
In Fig.3 A length bar will be better than the magnifier factor to appreciate the size of the red dots. In the experiment, the authors use the same amount of cDNA for all constructs. Why they do not double the amount of cDNA as in Fig. 2. Does the reduction of PLA is due to a decrease of the AGT expression protein? The authors should also specify how many times this experience has been done.
How many calls have been counted for each experience? It may help if the number of PLA was expressed by x number of cells. Furthermore, to support this result, the authors should have one image of each conditions that show the expression of AGT construct using HA antibody and anti-renin. Is it possible that mutant construct are less express in renin positive vesicles.
In Fig 4 The authors are talking about the effect of Hydrocortisone on stability. Should they write secretion or cytosolic density instead?
In Fig. 4B, the authors need to quantify the bands. The R375Q and 1-295 bands intensity seem to reduce in the presence of HC. The authors need to specify the number of performed experiments. In the legend, the authors should correct the 46h treatment performed in Hydrocortisone, not CHX.
Fig. 5 has not readable.
Does Fig. 6 show any not statistical difference? What about the reduction of signal in the wilt type plasmid below 40 ng? Should you use a dual luciferase assay?
Author Response
Reviewer 1.
Comments and Suggestions for Authors
The authors found a mutation in human angiotensinogen associated with the autosomal recessive renal tubular 30 dysgeneses (ARRTD). In this sequel, the authors study how this mutation plays a role in developing the ARRTD and show how Hydrocortisone can be beneficial.
Q1. In other to other groups to repeat the experiment, the authors had to give more details about their experimentation.
Response: We completely agree with this invaluable comment. As suggested, we have added detailed information about experimentation in the method.
Q2. How the authors verified the quality of the plasmids?
Response: Thanks for this important comment. Plasmid quality has been checked by GE NanoVue spectrophotometer (quotient A260/A280). A value of 1.80-1.90 is an indication for pure plasmid DNA. All plasmid DNAs were verified by sequencing analysis.
Q3. Where were the cells purchased? How long the plasmid was transfected, 12 hours, 24 hours?
Response: Thanks for these important comments. HK-2 and L02 cells were purchased from the Bioresource Collection and Research Center and Cell Bank of China Science (Shanghai, China), respectively. The plasmid was transfected for 46 hours.
Q4. What is the composition of the lysed buffer? Any protease inhibitor?
Response: The composition of the lysed buffer was 100 mM Tris-HCl (pH 8.0), 150 mM NaCl, 0.1% SDS and 1% Triton X-100. No protease inhibitor was added.
Q5. Where the antibodies were purchased, and their working concentration?
Response: Thanks for this question. Anti-Hemagglutinin monoclonal rat antibody (clone 3F10, Roche), 1:5000 dilution, anti-actinin mouse monoclonal antibody (H-2, sc-17829, Santa Cruz Biotechnology), 1:5000 dilution, and anti-AGT rabbit polyclonal antibody (11992-1-AP, Proteintech Group), 1:1000 dilution were used.
Q6. Why the authors describe coefficient 123 of variance (COV) when they do not use this parameter in the text?
Response: Thanks for this important comment. In fact, we did not use the COV. COV was deleted from the method.
Q7. In section 2.3, the authors wrote cycloheximide for 46 h, while in Fig.2, then use 0 to 3 hours.
Response: We greatly appreciate this important comment. This is our typo error. We have corrected as “the transfected cells were treated with 50 μg/ml cycloheximide (CHX) for the indicated time, such as from 0 to 3 hours for this analysis”.
Q8. In the PLA assay, the authors use a different cell culture medium described in the cell culture section.
Response: Thanks for this important comment. We have corrected as “HK-2 cells were cultured in DMEM/F-12 medium with 10% FBS, and L02 cells were cultured in DMEM medium with 10% FBS”.
Q9. Figure 1 is roughly the same as the previously published content. Is there any new information?
Response: Thanks for this important concern. As suggested, we have rearranged the figure and highlighted the candidate amino acid binding-motif of glucocorticoid receptor to avoid re-publication of the figure.
Q10. In Fig. 2 A, the authors show no bands in the absence of Wild type construct. However, a bar similar to the wt construct appears in both mutant constructs, suggesting endogenous AGT detection. Do the authors use HA antibody or AGT antibody, or both?
Response: Thanks for this critical comment. Current Figure 2A was detected with AGT antibody which could detect the endogenous AGT in wt and two mutant conditions. In addition, we also used HA antibody to verify the position of wt and two mutant proteins in gels (data not shown).
Q11. The authors suggest that HA-hAGT 1-292 is more stable than other constructs. We can imagine that the AGT reduction in the cells is mostly due to AGT being secreted from the cells because we do not observe any degradation products. Since the HA-hAGT 1-292 is less expressed than wild type constructs, it suggests a problem in folding in RE and Golgi that slow down its release in secret pathways. To confirm the AGT's degradation, authors should perform this experiment in the presence of protein degradation inhibitors.
Response: We extremely agree with this valuable comment. We have taken this comment into concern. Unlike missense mutations, the decreased protein expression may be involved in the pathway in RE and Golgi, leading to secretory defects (retention in RE or Golgi). Our homozygous E3_E4 del:2870bp deletion+9bp insertion in AGT is the frameshift mutation, which excludes the exon 3 and 4, creates the stop codon, and result in the generation of the truncated protein (AGT 1-292). The mechanism should be different. We found the reduced expression of HA-hAGT 1-292 and explored the stability of this truncated protein using the pulse-chase experiment by CHX as the classical method to study the stability of the protein. Due to the time limit, we are unable to complete the suggested experiment unless the editor strongly recommends it absolutely necessary.
Q13. In Fig.3 A length bar will be better than the magnifier factor to appreciate the size of the red dots. In the experiment, the authors use the same amount of cDNA for all constructs. Why they do not double the amount of cDNA as in Fig. 2. Does the reduction of PLA is due to a decrease of the AGT expression protein? The authors should also specify how many times this experience has been done.
Response: Thanks for these important comments. We have added a length bar in Figure 3A. The amount of plasmid DNAs for transfection failed to proportionally translate proteins, which had different sensitivity to AGT antibody. We have adjusted the amount of plasmid DNAs based on Figure 2A to perform the experiments of Figure 2B and Figure 3A. Hence, the reduction of PLA should not be caused by this reason. We have revised and added the information of repeat time in the revised manuscript.
Q14. How many cells have been counted for each experience? It may help if the number of PLA was expressed by x number of cells. Furthermore, to support this result, the authors should have one image of each conditions that show the expression of AGT construct using HA antibody and anti-renin. Is it possible that mutant construct is less express in renin positive vesicles.
Response: We greatly appreciate these important comments. As recommended, we have added the cell number used in PLA. The limitation of PLA analysis is to use two matched antibodies. Here, the current combination of rabbit anti renin antibody and mouse anti AGT antibody is the only choice.
Q15. In Fig 4 The authors are talking about the effect of Hydrocortisone on stability. Should they write secretion or cytosolic density instead?
Response: We agree with this suggestion. As suggested, we have rewritten as “the stabilization of cytosolic truncated AGT” in the revised manuscript.
Q16. In Fig. 4B, the authors need to quantify the bands. The R375Q and 1-295 bands intensity seem to reduce in the presence of HC. The authors need to specify the number of performed experiments. In the legend, the authors should correct the 46h treatment performed in Hydrocortisone, not CHX.
Response: Thanks for these important comments. We have added the number of performed experiments and corrected the 46h typo in this revised version. We also quantified the effect of HC on the band intensity of R275Q and 1-295 using the CHX pulse-chase experiment in Figure 4C. There is no specific information to quantify Figure 4B.
Q17. Fig. 5 has not readable.
Response: Thanks for this suggestion. We have revised and renewed the Fig. 5 for the readability.
Q18. Does Fig. 6 show any not statistical difference? What about the reduction of signal in the wilt type plasmid below 40 ng? Should you use a dual luciferase assay?
Response: Thanks for these important comments. As suggested, we have calculated the p value and also revised the Fig. 6 accordingly. The functional role of AGT proteins on transactivation is unsuitable to use dual luciferase assay. In general, we used total protein amount to normalize.
Reviewer 2 Report
This paper is about the use of hydrocortisone to treat the effect of a mutation on angiotensinogen.
I suggest checking all the paper for English language. There are some sentences that sounds weird.
The quality of the paper is appropriate, the references are new enough.
The quality of the figures should be improved. There are many results per figure, and the captions are not enough to describe the figure. Moreover, in the figure captions there is information that should be in the results section, not as figure captions.
There are some more specific comments, described below.
Line 34 Review this phrase, it is complicated to understand. “We have shown that a dose-dependent manner with a relatively low expression of this truncated AGT”
Line 51 Correct to “Almost all affected”
Lines 81-84. I suggest that the explanation of this figure should be in the text, not as figure caption
Line 132 there is a missing verb in “were in a dose-dependent manner”
Line 141. Review the figure caption, there is information that should be in text and not as figure caption. This observation applies to all figure captions. For example, the paragraph in lines 150-156 seems incomplete, and the information in figure 3 captions could complete this paragraph.
Figure 4. The graphs below Figure 4 C and D need more explanation.
Figure 5. review this graph, one part is missing, or is not correct
figure 7 check spelling of angiotensinogen
Author Response
Reviewer 2.
Comments and Suggestions for Authors
This paper is about the use of hydrocortisone to treat the effect of a mutation on angiotensinogen.
Q1. I suggest checking all the paper for English language. There are some sentences that sounds weird.
Response: Thanks for this suggestion. We have sent our manuscript for English editing by a native speaker.
Q2. The quality of the paper is appropriate, the references are new enough.
Response: Thanks for your positive comment.
Q3. The quality of the figures should be improved. There are many results per figure, and the captions are not enough to describe the figure. Moreover, in the figure captions, there is information that should be in the results section, not as figure captions.
Response: We greatly appreciate this valuable comment. As suggested, we have renewed captions of all figures and revised our results accordingly.
There are some more specific comments, described below.
Q4. Line 34 Review this phrase, it is complicated to understand. “We have shown that a dose-dependent manner with a relatively low expression of this truncated AGT”
Response: Thanks for this important concern. We have revised the sentence as “The expression of this truncated AGT proteins was relatively low with a dose-dependent manner.” for clarity.
Q5. Line 51 Correct to “Almost all affected”
Response: As suggested, we have corrected accordingly.
Q6. Lines 81-84. I suggest that the explanation of this figure should be in the text, not as figure caption
Response: Thanks for this important concern. As suggested, we have moved and revised the result and figure caption accordingly.
Q7. Line 141. Review the figure caption, there is information that should be in text and not as figure caption. This observation applies to all figure captions. For example, the paragraph in lines 150-156 seems incomplete, and the information in figure 3 captions could complete this paragraph.
Response: Thanks for this important concern. As suggested, we have revised the result and figure caption accordingly.
Q8. Figure 4. The graphs below Figure 4 C and D need more explanation.
Response: Thanks for this important suggestion. We have added more information regarding Fig. 4C and D.
Q9. Figure 5. review this graph, one part is missing, or is not correct
Response: Thanks for this important concern. We have revised and renewed Figure 5 for clarity.
Q10. Figure 7 check spelling of angiotensinogen
Response: We have corrected it.
Round 2
Reviewer 1 Report
he authors had answered most of the questions. However, it still one key question that the authors did not respond.
Authors based all the manuscript on the stability of the proteins using the pulse-chase experiment with CHX. To my knowledge, this is not a classical method to study the stability of protein as such as a secreted soluble protein. The lack of degradation product observed in the western blot suggested that angiotensinogen wt is not degraded but excrete. The authors have to show either the western-blot performed with HA antibody has a degradation band pattern or conduct an experiment with lysosomal or proteasomal degradation inhibitors to support their hypothesis. This is a critical experiment that has to be done to support the hypothesis of the manuscript.
In figure 3 the authors added in the legend the number of experiments performed. They did not write how many cells have been investigated. The data should be expressed in the number of PLA by cells since that in Fig 3B; we observed PLAs in none transfected cells. ( ex. HA Vector, 3 PLA by 100 cells, 1-485 (56 PLA by 100 cells)). From the picture, the abondance of the AGT is only 3 times higher than the vector alone. Figure 5 has 10 times more PLA in control than the control of figure 3. The expression by 100 cells should give a more accurate value. In our laboratory, we take more than 300 cells by a condition in a double-blind manner. This is a repeat at least 3 times. Since the authors already had all data. It should not take more than a day to quantify.
In figure 4, the authors had to show the quantification of all A, B, C, and D panels. The authors should explain why the treated mutant bands are significantly reduced in 4B in the presence of HC if these bands have lower intensities, why this effect is not visible in the panel 4C and D.
In legend 4, the authors suggest that wt and both HA.hAGT mutants are increased by HC. However, graph 4C shown no significant difference between the vehicle and the HC treated cells.
The authors added information in the latest version. However, they forget few elements that they provide to the referee. Please add this information to the text.
1)The author has to write in material and method how the plasmid has been confirmed. “All plasmid DNAs were verified by sequencing analysis using….method, core”.
2) The authors answered that the plasmid was transfected for 46 hours. However, at lane 142, they wrote 36h.
Reviewer 2 Report
Your paper has improved greatly.
There are only two details about grammar:
Line 99. I suggest writingL the time instead of “the indicated time”
Line 154 I suggest using “relative” instead of “relatively”.
Author Response
Reviewer 2.
Comments and Suggestions for Authors
Your paper has improved greatly.
There are only two details about grammar:
Q1. Line 99. I suggest writing the time instead of “the indicated time”
Response: Thanks for your suggestion. We have replaced the indicated time with actual time (20, 40, 80, 180, and 360 minutes) in the text.
Q2. Line 154 I suggest using “relative” instead of “relatively”.
Response: We have corrected this typo.
Round 3
Reviewer 1 Report
The hypothesis is based on the idea that the protein is degraded inside the cells. But, it is known that the protein is secreted. It has two possibilities to interpret the results. The authors lack an important control experiment. This control is a routine experiment, and it should not be difficult to perform. The authors have to perform a cycloheximide chase assay in the presence of degradation inhibitors. They have to show that the proteins should not disappear or is partially reduced.
Author Response
We showed the rescued data by a proteasomal degradation inhibitor MG132 in the Western-blot performed with HA antibody to support our hypothesis in this revised version.
Round 4
Reviewer 1 Report
The authors have anwered the question.